# Antimicrobial peptides in frog poisons constitute a molecular toxin delivery system against predators

Constantijn Raaymakers[1,2], Elin Verbrugghe[2], Sophie Hernot[3], Tom Hellebuyck[2], Cecilia Betti[4], Cindy Peleman[3], Myriam Claeys[5], Wim Bert[5], Vicky Caveliers[3], Steven Ballet[4], An Martel [ID] [2], Frank Pasmans[2] & Kim Roelants[1]

Animals using toxic peptides and proteins for predation or defense typically depend on specialized morphological structures, like fangs, spines, or a stinger, for effective intoxication. Here we show that amphibian poisons instead incorporate their own molecular system for toxin delivery to attacking predators. Skin-secreted peptides, generally considered part of the amphibian immune system, permeabilize oral epithelial tissue and enable fast access of cosecreted toxins to the predator's bloodstream and organs. This absorption-enhancing system exists in at least three distantly related frog lineages and is likely to be a widespread adaptation, determining the outcome of predator–prey encounters in hundreds of species.

[1] Amphibian Evolution Lab, Biology Department, Vrije Universiteit Brussel, Pleinlaan 2, 1050 Elsene, Belgium. [2] Department of Pathology, Bacteriology and Avian Diseases, Faculty of Veterinary Medicine, Ghent University, Salisburylaan 133, 9820 Merelbeke, Belgium. [3] Department of Nuclear Medicine, UZ Brussel and In vivo Cellular and Molecular Imaging, Vrije Universiteit Brussel, Laarbeeklaan 103, 1090 Jette, Belgium. [4] Research Group of Organic Chemistry, Department of Chemistry and Department of Bio-engineering Sciences, Vrije Universiteit Brussel, Pleinlaan 2, 1050 Elsene, Belgium. [5] Department of Biology, Nematology Research Unit, Faculty of Science, Ghent University, 9000 Ghent, Belgium. Correspondence and requests for materials should be addressed to K.R. (email: Kim.Roelants@vub.be)

When a poisonous or venomous animal is attacked by a predator, it usually has little time to deploy its toxins and induce a response in the aggressor to avoid being killed (Fig. 1a). To achieve this, animals may produce toxins with immediate local effects, like pain or distastefulness, or molecules with systemic effects that rely on fast infiltration into a predator's body[1]. In the absence of structures that create a wound to inject toxins, fast delivery is typically only possible for small molecules (<300 Da) that are easily absorbed through oral epithelia, like cyanides, alkaloids, or steroids[1,2]. Many frog species however, secrete much larger molecules with systemic targets, like peptides (typically 0.5–2 kDa) and in some cases even small proteins (4–8 kDa)[3–7]. These toxins generally resemble vertebrate hormones or neuropeptides and undergo posttranslational modifications that increase their lifespan in a predator's bloodstream or enhance their affinity to a target receptor[8–10]. Upon receptor binding, these toxins essentially simulate an overdose of a predator's hormone or neuropeptide, resulting in a range of adverse effects, including nausea, hypotension, and hyperalgesia[3,7,9,10].

The large size and physicochemical properties of peptides renders them unsuitable for fast epithelial absorption[1,2,11–13]. The poisons of many frogs thereby seem to defy the constraints of passive toxin uptake, and hint at a mechanism that allows systemic delivery fast enough to evade predation. Besides toxins, many frogs secrete antimicrobial peptides (AMPs), capable of killing a broad range of microorganisms through cell lysis[14,15]. Since their first characterization in the late 1980s, AMPs have been generally considered a component of the amphibian innate immune system, and investigated primarily in light of their potency against clinically important pathogens[3,14–18]. Recently however, an alternative role in antipredator defense has been suggested, and it was predicted that AMPs, through a similar cytolytic activity, could permeabilize a predator's epithelial tissue to facilitate toxin delivery[7]. Proof for this hypothesis would highlight a long-overlooked function of amphibian AMPs besides innate immunity, and drastically alter our perception of how they contribute to an amphibian's survival. Here we report on a series of in vitro and in vivo experiments that provide compelling evidence for an antipredatory role of AMPs. Our results complete the picture of a toxin delivery system as sophisticated as that of many venomous animals, be it molecular rather that morphological in nature.

## Results

**AMP-enhanced transepithelial passage of a peptide toxin.** As the main model for our study, we used a peptide pair composed of the systemic toxin caerulein, and the AMP caerulein precursor

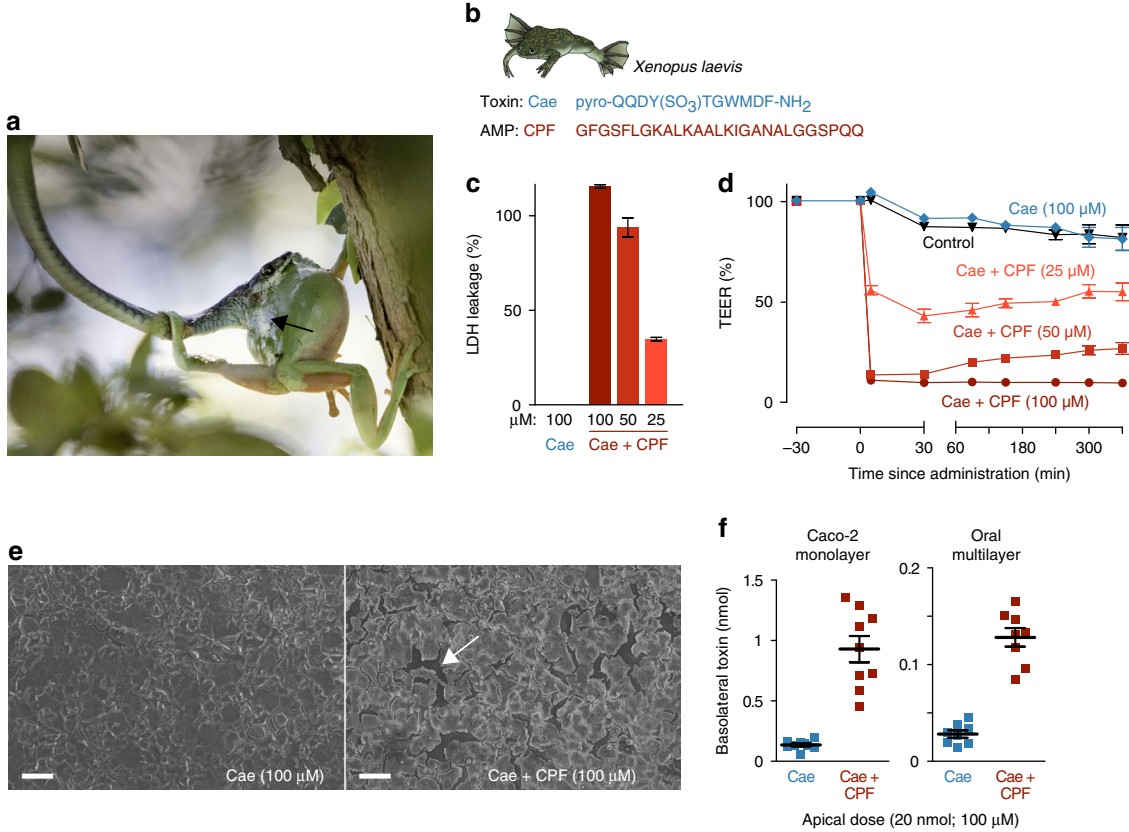

**Fig. 1** A frog antimicrobial peptide promotes the transepithelial passage of a cosecreted toxin. **a** *Litoria caerulea* (an Australian tree frog) attacked by the snake *Dendrelaphis punctulatus*. Note the defensive skin secretion (black arrow), known to contain toxins (including caerulein) and AMPs (photograph taken by Jannie Smit). **b** Amino-acid sequences of the *X. laevis* skin toxin caerulein (Cae) and its cosecreted antimicrobial peptide, caerulein precursor fragment (CPF). **c** Lactate dehydrogenase (LDH) leakage indicates cell damage in a Caco-2 epithelial model exposed to a mixture of caerulein and CPF but not to caerulein alone ($n = 7$; 100% corresponds to the LDH leakage caused by complete cell lysis as induced by Triton-X). **d** Co-administration of CPF induces a rapid prolonged drop in transepithelial electrical resistance (TEER) of Caco-2 monolayers ($n = 9$, one-way ANOVA, $F_{(4, 40)} = 1345.6$, $p < 0.0005$). **e** Caerulein alone does not damage Caco-2 monolayers, but co-administration of CPF results in intercellular ruptures (white arrow), as revealed by scanning electron microscopy. Scale bars represent 100 μm. **f** Co-administration of CPF at the apical side leads to higher caerulein levels after 60 min at the basolateral side, indicating enhanced transepithelial transport across Caco-2 monolayers ($n = 9$, t-test, $t_{(8.2)} = -7.3$, $p < 0.0005$) and an oral multilayer model ($n = 8$, t-tests, $t_{(9.1)} = -9.6$, $p < 0.0005$). All data are mean ± s.e.m., error bars not shown when covered by data symbols

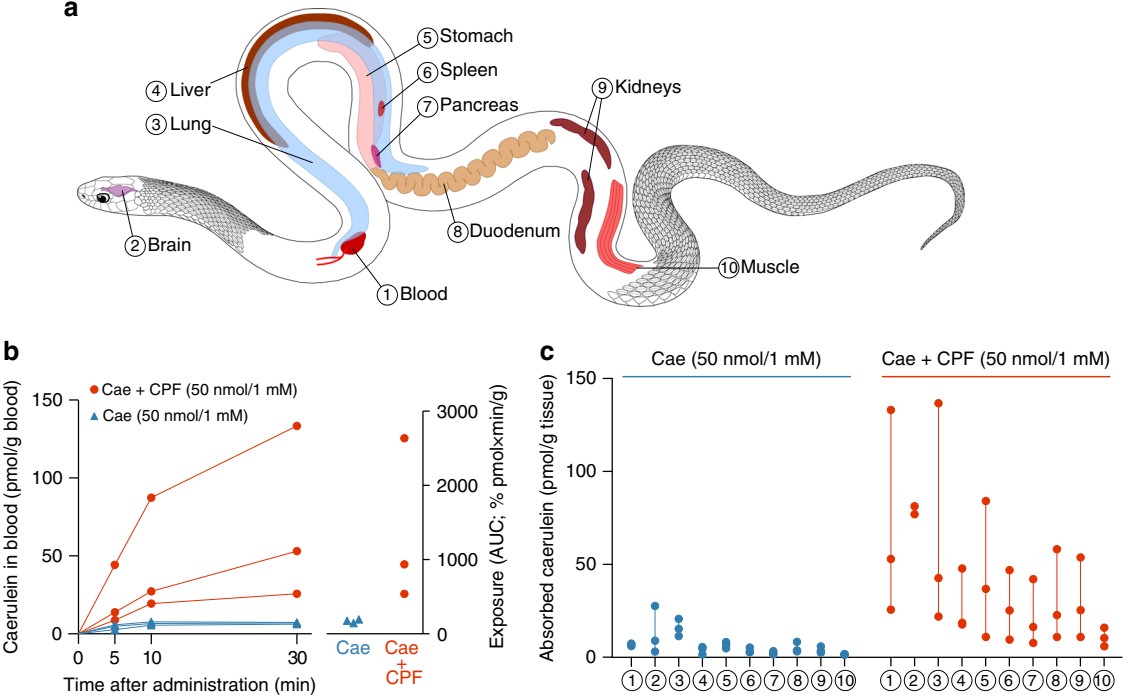

**Fig. 2** AMP co-administration accelerates toxin absorption in a live predator. **a** Investigated organs and tissues in the model predator *Thamnophis eques*; numbers are cross-referenced in **b** and **c**. **b** Caerulein (Cae) blood concentrations reach higher levels (left graph), which leads to higher systemic exposure (right graph) when orally co-administered with CPF at 1 mM ($n = 3$, linear mixed model, $X^2$ (1) = 11.4, $p < 0.001$). **c** Caerulein levels in organs are higher in snakes when co-administered with CPF after 30 min ($n = 3$, linear mixed model, $X^2$ (1) = 4.7, $p = 0.029$). Symbols represent individual data values. Linear mixed models are explained in the Methods section

fragment-3 (CPF), both of which are key constituents of the skin secretion of the frog *Xenopus laevis*[18–20] (Fig. 1b). Caerulein is a potent ligand of cholecystokinin receptors expressed in the vertebrate intestine, pancreas, and brain[9], and at sufficiently high concentrations, may induce nausea, hypotension, intestinal cramps, vomiting, and diarrhea[9,21,22]. Even in the intestine, the target receptors are only accessible by means of epithelial absorption, since they are expressed at the basolateral side (inside the tissue) of the intestinal wall. Because caerulein is a widespread toxin in frogs[3,7,9] with a long half-life in the bloodstream (up to 45 min)[8,23], this peptide makes a suitable model to investigate mechanisms of toxin delivery.

The absence of lactate dehydrogenese (LDH) leakage after applying caerulein (100 μM) to epithelial cell cultures indicate that the toxin is incapable to damage epithelial cells by itself (Fig. 1c). However, co-administration of caerulein and CPF, resembling the natural condition of the frog's skin secretion, resulted in dose-dependent LDH leakage within 5 min indicating rapid and large-scale damage to cell membranes. Application of CPF alone produced a similar level of LDH leakage, confirming that this peptide, besides lysing bacteria, is capable to perforate epithelial cells (Supplementary Fig. 1a). Accordingly, administration of CPF and caerulein combined, or CPF alone, strongly reduced epithelial barrier integrity as measured by transepithelial electrical resistance (TEER) across cell monolayers (Fig. 1d; Supplementary Fig. 1b). While application of caerulein alone caused no apparent loss of TEER, exposure to CPF caused an immediate, dose-dependent drop in epithelial barrier integrity. At 100 μM caerulein + 100 μM CPF, TEER dropped to 10% ± 2.5 of the original value within 5 min. The loss of epithelial integrity is explained by damage to the monolayer that exceeds the mere lysis of individual cells (Fig. 1e). The presence of CPF caused abundant intercellular ruptures in the monolayer, indicating that adherent

cells were detached, possibly as a consequence of CPF-induced cytolysis. This damage allowed the passage of caerulein across cell monolayers as confirmed by enzyme-linked immunosorbent assay (ELISA; Fig. 1f). When caerulein alone (100 μM or 20 nmol) was applied at the apical side of the monolayer (representing the oral cavity), ~ 0.3–1.0% (0.06–0.20 nmol) of the toxin had passed to the basolateral side (representing a predator's tissue and blood) after 60 min. When CPF was co-administered (at the same concentration), transepithelial passage of caerulein rose to 2.3–6.8% (0.45–1.36 nmol) of the applied dose, representing a near sevenfold increase on average in the rate of transepithelial passage. Similarly, a nearly fivefold increase in toxin passage was observed when an oral epithelial model composed of 8–11 stacked cell layers was used, showing the capacity of the AMP to enhance peptide passage through thicker, more complex epithelia (Fig. 1f). Although for both epithelial models, the percentage of transferred caerulein remains relatively low, the significant increase caused by the AMP is likely to provide a major advantage to a frog under attack, provided that its skin secretion contains a high concentration of the toxin (see next section).

**AMP-enhanced intoxication of a live predator**. Our in vitro experiments provide a strong indication that frog AMPs indeed enhance the transepithelial passage of cosecreted toxins. To test whether this pattern holds in a live model predator, we quantified radiolabeled caerulein (Supplementary Fig. 2) absorbed by snakes until 30 min after oral administration in the presence vs. absence of CPF (Fig. 2a). This time window corresponds well with reported times required for snakes and several other predators to subdue and ingest frogs[24,25]. Despite the long half-life of caerulein, radioactivity measurements across such period may increasingly overestimate the amount of intact toxin in a

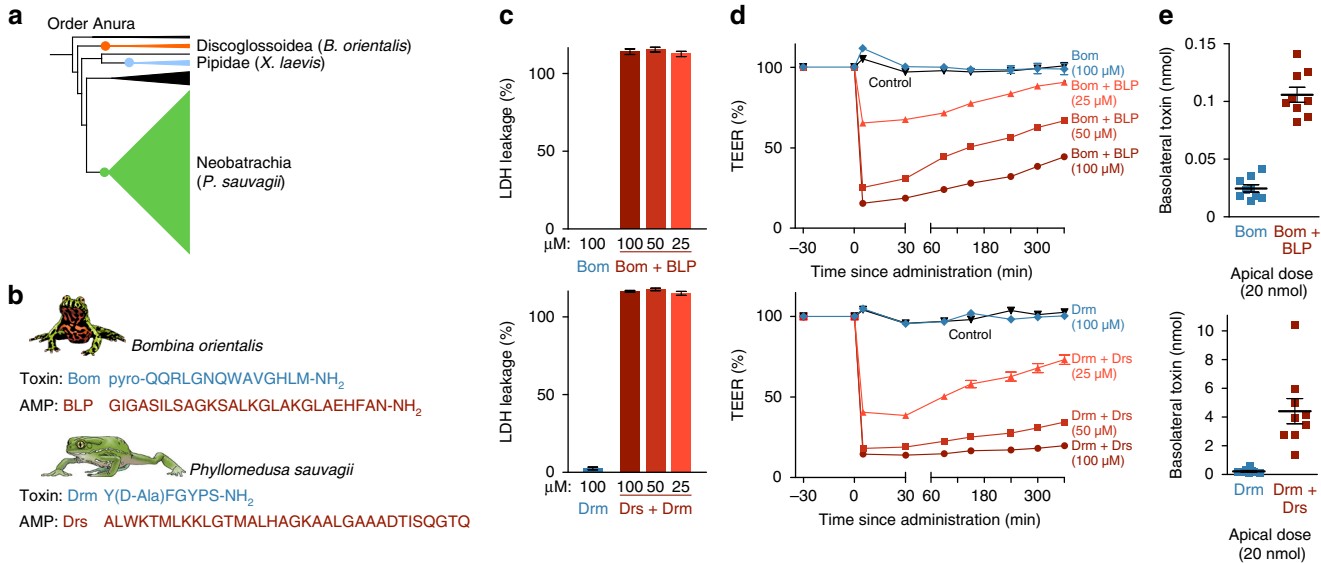

**Fig. 3** The role of frog AMPs in toxin absorption in other frog species. **a** Representatives of three major frog lineages cosecrete peptide toxins and AMPs. **b** Amino-acid sequences of the *Bombina ortienalis* toxin bombesin (Bom) and AMP bombinin-like peptide 1 (BLP), and of the *Phyllomedusa sauvagii* toxin dermorphin (Drm) and its AMP dermaseptin-S1 (Drs). **c** LDH leakage indicates cell damage in Caco-2 monolayers when exposed to the AMP + toxin mixtures, but not to the toxins alone ($n = 6$–7; 100% corresponds to the LDH leakage caused by complete cell lysis as induced by Triton-X). **d** BLP and Drs induce a dose-dependent rapid drop in transepithelial electrical resistance (TEER) of Caco-2 monolayers ($n = 9$, one-way ANOVA, $F_{DRS}(4,40) = 1284$, $F_{BLP}(4,40) = 2054.1$, $p < 0.0005$). **e** Co-administration of BLP and Drs at the apical side of Caco-2 monolayers leads to higher toxin levels after 60 min at the basolateral side, showing enhanced transepithelial transport of bombesin ($n = 9$, t-test, $t(11.6) = -11.2$, $p < 0.0005$) and dermorphin ($n = 8$–9, t-test, $t(8.1) = -4.8$, $p < 0.0005$). All data are mean ± s.e.m., error bars not shown when covered by data symbols

predator's body. Nevertheless, the radiolabeled peptide provides a useful tool to quantify and compare the efficiency of epithelial absorption as the first step of predator intoxication.

In a first in vivo experiment, we administered both peptides at 1 mM (50 nmol in 50 μl buffer solution). This is a biologically realistic concentration for frog skin secretions according to reported estimates for AMPs[26–28] and ELISA estimates for caerulein (0.12–4.27 mg caerulein per frog, corresponding to 0.088–3.16 μmol, or 1.0–7.5 mM in the skin secretions; Supplementary Fig. 3). Monitoring of caerulein levels in the snakes' blood after oral administration revealed a variable but strong effect of the AMP on toxin absorption (Fig. 2b). In the absence of CPF, caerulein blood levels remain low (below 8 pmol/g blood) throughout the experiment, but in the presence of CPF, they reach concentrations of 19–87 pmol/g blood after 10 min and 26–133 pmol/g blood after 30 min. Taking into account that a snake's blood takes 5–8% of its body weight (BW)[29], these blood levels correspond to 0.96–4.36 pmol/g BW after 10 min and 1.28–6.66 pmol/g BW after 30 min. The dose at which caerulein causes adverse effects in snakes is unknown, but the levels observed here largely exceed intravenously injected doses that in previous experiments caused severe sickness in other vertebrates (1 μg/kg BW, corresponding to 0.74 pmol/g BW)[9,21,22]. Over 30 min, systemic exposure to the toxin (area under the curve (AUC) derived from repeated blood measurements) ranged up to 13 times higher in the presence of CPF (Fig. 2b), and average caerulein levels in various organs ranged between 1.3 and 10.4 times higher (Fig. 2a, c). The higher caerulein levels in all organs suggest that the AMP did not direct the toxin to its specific target organs (intestine, pancreas, and perhaps the brain), but instead facilitated dispersal of the toxin throughout the predator's system by accelerating absorption. A second in vivo experiment, based on the oral administration of 10-fold lower peptide concentrations (100 μM) confirmed that the effect of the AMP is still apparent at the concentration used in our in vitro experiments. In the presence of CPF, estimated exposure was up to three times

higher and average organ concentrations after 30 min ranged between 1.7 and 3.4 times higher (Supplementary Fig. 4).

**A widespread role of frog AMPs in toxin absorption.** AMPs have been characterized in hundreds of frog species, and have been postulated to represent independent evolutionary origins in distantly related frog lineages[30]. Besides CPF and related peptides in species of the family Pipidae[18–20], AMPs are particularly prominent in the skin secretions of discoglossoid and neobatrachian frogs, in which they, too, co-occur with peptide toxins (Fig. 3a). To investigate whether absorption-enhancing capacity is a shared trait of AMPs from phylogenetically distant frog lineages, we conducted experiments on two additional peptide pairs (Fig. 3b): The toxin bombesin (Bom) and the AMP bombinin-like peptide 1 (BLP) from the skin of the oriental fire-bellied toad *Bombina orientalis* (Discoglossoidea: Bombinatoridae); and the toxin dermorphin and the AMP dermaseptin-S1 (Drs) from the skin of the waxy tree frog *Phyllomedusa sauvagii* (Neobatrachia: Hylidae). The toxin bombesin is a potent ligand of neuromedin receptors expressed on smooth muscle cells surrounding the gut and blood vessels, which causes hypotension and smooth muscle spasms[31–33]. Dermorphin causes sedation and shock upon binding its target opioid receptors in the nervous system[10,34,35]. Like caerulein, both toxins require access to the bloodstream to induce these effects, and have structural adaptations to extend their half-lives in the bloodstream and tissues[34,36].

Similar to caerulein, neither bombesin nor dermorphin induced any damage to epithelial tissue by themselves (Fig. 3c; Supplementary Fig. 5a). In contrast, co-administration of their respective AMPs, or application of the AMPs alone resulted in rapid and severe cell leakage and loss of epithelial integrity (Fig. 3d; Supplementary Fig. 5b). BLP and Drs enhanced the transepithelial passage of their cosecreted peptide toxins by ~ 4.5 and 20 times, respectively (Fig. 3e). These results suggest that absorption-enhancing activity is likely to be a prevalent feature of

anuran AMPs, contributing to predator intoxication by a wide range of species.

## Discussion

Most research on amphibian skin-secreted peptides has a predominantly pharmacological scope, aiming to characterize their physiological effects, not their natural mode of delivery. By investigating the process of intoxication "from a frog's perspective", our study has two major implications.

First, we provide evidence for a function of amphibian AMPs outside the immune system. Due to their pore-forming activity and expression in the skin, AMPs are inclined to serve a dual role, in innate immunity as well as antipredator defense. In fact, besides enhancing the absorption of cosecreted toxins, the cytotoxic activity of amphibian AMPs may cause irritation or pain on oral mucosa, thereby providing a secondary and fast-acting effect on some predators. The broad-spectrum antimicrobial activity of many amphibian AMPs continues to fuel the perception that they are promising lead compounds for the development of new antibiotics[3,18,37,38]. The present evidence however, implies that pharmacological studies seeking to optimize the therapeutic potential of frog AMPs should take their potential damaging effect on epithelial tissue into account.

Second, our results show that toxin systems relying on epithelial absorption may be more complex than currently perceived, involving multiple factors that together maximize the effectiveness of intoxication. At least in peptide-secreting frogs, epithelial permeabilization is likely to be a decisive step, reducing the time for toxins to reach an adverse effect in the predator. The secretion of large amounts of toxin additionally increases the probability that effective toxin levels are attained in the predator's system even if only a small percentage of the toxin is absorbed. Similar to the large amounts of caerulein in *X. laevis* skin secretion found here, quantities ranging up to milligrams have been reported for toxins in the skin secretion of other frog species, including caerulein in *Litoria caerulea* (Fig. 1a), and bombesin and dermorphin in *Bombina* and *Phyllomedusa* species, respectively[10,39,40]. With molecular weights of, respectively, 802.9, 1352.4, and 1619.9 Da, dermorphin, caerulein, and bombesin lie within the typical size range of anuran peptide toxins. A few toxins however, like anntoxin in *Hyla annectans*, sauvagine in *P. sauvagii*, or prokineticins in *Bombina* are much larger (4–8 kDa)[4–6]. The question remains whether toxins of this size would also benefit from AMP-enhanced absorption. The degree by which epithelial transfer was enhanced in our in vitro experiments seemed inversely correlated to the toxins' molecular weights, which may indeed indicate that a peptide's size affects an AMP's capacity to mediate its transepithelial uptake.

Absorption enhancement is likely to be effective against predators whose limited prey-handling skills require considerable time to subdue their prey, and/or swallow their prey whole (e.g., predatory fish, lizards, and large amphibians). In addition, it may act in concert with ingestion-delaying behavior such as struggling or inflating. For various snake species, recorded times between capture and ingestion of amphibians averaged 15–35 min[24] and in some cases ranged up to 50 min[25]. Similarly, amphibians have been documented to survive up to 20 min in the gastrointestinal tract of bullfrogs and fish after being swallowed completely[25]. If in the latter case sickness is induced before the frog dies, it might be regurgitated and escape. Even in predators whose preying technique precludes survival of the frog (e.g., involving severe trauma before ingestion), enhanced toxin absorption could still adversely affect the predator and be beneficial to the remaining frog population, possibly aided by predator learning[25,41,42].

We anticipate that other animals administering their toxins through absorption may use similar mechanisms of tissue permeabilization. The defense secretions of several whipscorpion species (order Thelyphonida), for example, contain caprylic acid, which, besides increasing the effective contact area of the sprayed secretion, enhances the penetration of cosecreted acetic acid through arthropod cuticle[43]. Examples involving toxic peptides or proteins with systemic targets may include salamanders[44], nemertean worms[45], and cone snails releasing insulin-based secretions[46]. Moreover, the venoms of animals relying on active injection like biting or stinging (spiders, insects, and cnidarians) often include cytolytic agents, which may further improve toxin spreading[47–49]. As such, our findings call for a rethinking of the textbook distinction between poisonous animals (relying on absorption of their toxins) and venomous ones (creating a wound to guide toxins into the victim)[1,2,50,51]. Well-known toxin delivery systems range from macroscopic structures (like snake fangs or insect stingers) down to cellular adaptations (nematocysts in cnidarians). The molecular delivery system identified here represents an extension down the biological scale, and show that toxin delivery can be mediated at any suborganismal level.

## Methods

**Ethics statement.** Experiments involving live animals were conducted in accordance with the European guidelines and the Belgian legislation on animal housing and experimentation, and were approved by the Ethical Committees of Animal Experimentation of the Vrije Universiteit Brussel (Permit nos. EC15-272-7 and EC16-334-1) and by the Ethical Committee of the Faculty of Veterinary Sciences of Ghent University (Permit no. EC2015/26).

**Choice and synthesis of toxins and AMPs.** Three toxin–AMP pairs were selected as models for this study, each representing cosecreted peptides from different clade of frogs: (1) the toxin caerulein (Cae) and the AMP caerulein precursor factor-3 (CPF) from the skin of the African clawed frog *X. laevis* (Pipidae); (2) the toxin bombesin (Bom) and the AMP BLP from the skin of the oriental fire-bellied toad *B. orientalis* (Discoglossoidea: Bombinatoridae); and (3) the toxin dermorphin (Drm) and the AMP dermaseptin-S1 (Drs) from the skin of the waxy tree frog *P. sauvagii* (Neobatrachia: Hylidae). Sequences and molecular weights of these peptides are provided in the appropriate figures (Figs. 1 and 3). The toxins were selected based on their well-documented systemic effects in vertebrates[9,10,31] and on the commercial availability of ELISA kits to allow their in vitro quantification (see below). The selected AMPs show broad-spectrum antimicrobial activity as confirmed by previous studies[18,52,53] and are considered part of the frogs' innate immune system[3,14–18]. All peptides were synthesized via standard Fmoc-based solid-phase peptide synthesis by the authors and by CASLO Laboratory ApS (Lyngby, Denmark). For radiolabeling, a caerulein analog, modified with a diethylenetriaminepentaacetic acid (CHX-A″-DTPA) at the N-terminal side of the peptide (i.e., CHX-A″-DTPA-βAla-Pro-Gln-Asp-Tyr-Thr-Gly-Trp-Met-Asp-Phe-NH2), as a chelator for the isotope indium-111 (Supplementary Fig. 2), was synthesized. Resin-bound peptide (Fmoc-βAla-Pro-Gln(Trt)-Asp(OtBu)-Tyr (OtBu)-Thr(OtBu)-Gly-Trp(Boc)-Met-Asp(OtBu)-Phe-Rink Amide resin) was synthesized manually and after Fmoc deprotection of the N-terminal β-Ala residue, a solution of CHX-A″-DTPA•3HCl (1.4 eq.) and DIPEA (9 eq) in dichloromethane (DCM)/dimethylformamide (DMF) (2:1, v/v) was added to the resin-bound peptide and the reaction mixture was stirred at room temperature for 16 h. The resin was washed subsequently with DMF, 2-propanol, and DCM. The complete conversion was verified by Kaiser test, as well as by high-performance liquid chromatography (HPLC) analysis. The cleavage of the peptides from the resin and the removal of acid-labile protecting groups were achieved using a mixture of TFA/TES/H2O (95:2.5:2.5, v/v) for 3 h. After evaporation of the cleavage mixture and precipitation in cold Et2O, the crude peptides were purified by semi-preparative reverse phase-HPLC using a mixture of CH3CN/H2O, containing 1% trifluoroacetic acid (TFA), as mobile phase. The peptides were delivered as HPLC-purified ( > 95%) TFA salts. The caerulein-modified peptide containing (CHX-A″-DTPA) was isolated with 7% yield. Most peptide salts were dissolved in Milli-Q water to obtain 10 mM peptide stock solutions for storage at 4 °C.

**Cell culturing.** Two models of gastrointestinal epithelia were selected to represent the mucosal barrier that toxins must cross in order to enter the basolateral tissue and blood of a predator. The Caco-2 cell line (Human cancerous intestinal cells, authenticated using light microscopy) was chosen because of its wide application in in vitro epithelial drug transport studies. EpiOral 3D tissues (ordered from Mattek TM, USA) were selected to test toxin transport across a more complex epithelial model composed of 8–11 cell layers. Caco-2 cells were cultured in cell medium (87% Dulbecco's modified Eagle's medium nutrient mix, 10% fetal calf serum,

1% non-essential amino acid, 1% kanamycin, 1% penicillin–streptomycin, and 0.1% amphotericin B) in a humidified incubator (90% humidity, 5% $CO_2$) at a constant temperature of 37 °C. To prepare an experiment, cultured cells were seeded on the appropriate inserts/wells/coverslips 21 days in advance, at a concentration of $2 \times 10^5$ cells/ml and further cultured to develop confluent and fully differentiated monolayers (composed cells with polarized apical and basolateral membranes and cell-cell adhesion complexes). The progress of confluence and differentiation was checked by light microscopy and measuring electrical resistance across the cell layer, respectively.

**LDH leakage assays.** Cell-damaging activity of the peptides was tested using commercial LDH II assay kits (Abcam, USA) and Caco-2 cell cultures prepared in 96-well plates (see above), with each well containing 200 μl of medium. For each toxin–AMP pair (see above), cell cultures were exposed for 5 min to three different AMP + toxin concentrations (100, 50, and 25 μM) and a single toxin concentration (100 μM). Additionally, we tested the effect of only administering the AMPs at the same three concentrations. Nine wells containing cells and medium but no peptide served as negative controls (representing background LDH release in the absence of cell damage, or "0% LDH leakage"). Six wells containing 1% triton-X lysis solution were used as positive controls (representing the LDH release caused by 100% cell lysis, or "100% LDH leakage"). After completing the colorimetric reaction according to the kit's manual, optical density (OD) was measured at 450 nm using a multiskan FC microplate photometer/ELISA reader (Thermo Scientific). The cell-damaging effect of a peptide (in % LDH leakage) was calculated as: $100 \times (OD_X - OD_N)/(OD_P - OD_N)$, where $OD_X$ is the corrected OD value of sample x, $OD_N$ is the average corrected OD obtained for the nine negative control wells, and $OD_P$ is the average corrected OD obtained for the six positive control wells. Note that the resulting % LDH leakage can be higher than 100%, e.g., if a peptide does not lyse all cells (unlike the total cell lysis in our positive control), leaving the surviving and damaged cells to continue to produce and release additional LDH. Consequently, our LDH leakage assays should be regarded as measures of cell membrane damage, not cell death.

**Monitoring of epithelial permeability.** The effect of peptide administration on epithelial permeability was investigated by monitoring TEER on Caco-2 cell monolayer cultures. For each toxin–AMP pair (see above), five treatments ($n = 9$) were monitored: (1) apical administration of the toxin alone, at 100 μM; (2) apical administration of toxin + AMP, both at 100 μM; (3) toxin + AMP, both at 50 μM; (4) toxin + AMP, both at 25 μM; and (5) no peptide administration (negative control). Caco-2 cells were prepared (see above) on collagen-coated polystyrene transwell insert filters with a 0.4 μm pore size (Corning Life Sciences, USA). Inserts were placed in 24-well plates (provided with the inserts), creating an apical compartment (the insert) and a basolateral compartment (the well). Monitoring of TEER was done using a STX2 "Chopstick" electrode connected to an EVOM voltohmmeter (World Precision Instruments Inc). The original TEER (TEER$_O$; corresponding 100% epithelial integrity), was measured 30 min before administration of the peptides. Peptides were administrated by replacing the apical medium by fresh medium in which the peptides were dissolved. As frequent exposure to electrical currents can damage epithelial cells (explaining the shallow drop of TEER at time 0 in the negative controls in Figs. 1d and 3d), an incubation time of minimum 30 min between successive TEER measurements was respected. Hence, TEER was measured 5 min after peptide administration and then at 30-min intervals until 3 h after administration, and at 60-min intervals until 6 h after administration. A final measurement was taken at 24 h after peptide administration to monitor recovery of the monolayers after the peptide treatment (data not shown). Relative TEER (in %) values were calculated as $100 \times TEER_X/TEER_O$, where TEER$_X$ is the measured TEER at time X.

**Scanning electron microscopy imaging.** Caco-2 cells were seeded on glass coverslips that were coated with rat tail collagen 1 (Sigma-Aldrich) the day before. The effect of four treatments (100 μM CPF, 100 μM caerulein, 100 μM CPF + caerulein, and a negative control) on differentiated Caco-2 monolayers was visualized. Chemical fixation was performed in 12-well culture plates 5 min after peptide administration (700 μl of solution in all treatments). For fixation, plate wells were filled with 1 ml 2,5% glutaraldehyde in 0.05 M sodium cacodylate buffer (pH 7.4; 37 °C) for 20 min, and subsequently washed twice with the same buffer. Postfixation was performed with 1% osmium tetroxide for 30 min, followed by three rinses with double-distilled water. Dehydration was carried out using a graded hexylene glycol series: 30; 50; 70; and 90%, taking 15 min for each step. Cells were rinsed three times for 10 min with 100% hexylene glycol, and then two more times for 10 min with 100% ethanol. Critical-point drying of coverslips was done using a Balzers union CPD020. Dried coverslips were mounted on carbon-coated stubs with double-sided conductive tape. Gold coating happened with a Jeol 1200JFC fine coater, after which coverslips were examined with a Jeol JSM-840 scanning electron microscope operating at 12 kV.

**Quantification of transepithelial toxin passage.** Toxin concentrations were quantified by ELISA using commercially available kits that show 100% cross-reactivity with the toxin in question. Caerulein was quantified using the

Cholecystokinin Octapeptide (CCK) (26-33) (non-sulfated) ELISA kit (Phoenix Pharmaceuticals Inc., Burlingame, USA), bombesin was quantified using the Bombesin ELISA kit (Peninsula Laboratories International Inc.), and dermorphin was quantified using the MaxSignal Dermorphin ELISA kit (Bioscientific.com). For each toxin–AMP pair, AMP-induced increase in the passage of the toxin across Caco-2 monolayers was investigated by comparing basolateral concentrations at 60 min after apical administration of either 20 nmol of the toxin alone (yielding a concentration of 100 μM; $n = 9$), or after apical administration of 20 nmol toxin + 20 nmol AMP (100 μM for both; $n = 9$). Toxin passage across EpiOral multilayered cell cultures was investigated for the caerulein–CPF pair in a similar setup ($n = 8$). Each basolateral sample was stepwise diluted in cell medium and transferred to an ELISA kit plate. Six wells containing negative control samples (only cell medium) and two wells containing positive control samples (supplied with the kits) were added. Assays were performed following the standard protocols provided with the kits' manuals. Standard curves for each toxin were obtained in duplicate. OD was measured at a wavelength of 450 nm using a multiskan FC microplate photometer/ELISA reader (Thermo Scientific). OD values were converted into toxin concentrations using the four-parameter curve fitting software (Masterplex, MiraiBio Group, Hitachi Solutions America Ltd and elisa-analysis.com).

**Quantification of peptides in skin secretion.** Previous studies have estimated that AMP concentrations in the skin secretion of various frog species are in the millimolar range[26–28]. For X. laevis, AMP concentration estimates range between 4.5 and 19.6 mg/ml, and CPF seems one of the most abundant peptides[18–20]. Using CPF as a proxy for all X. laevis AMPs, and given its molecular weight of 2602 Da, these observations would correspond to a CPF concentration of ~1.8–7.8 mM. The lower limit of this range is slightly above the concentration used in part of our in vivo experiments (see below). Caerulein concentrations in X. laevis skin secretions were estimated using ELISA. Adult female X. laevis frogs ($n = 8$) were ordered from a licensed commercial breeder (Xenopus Express France) and housed in plastic containers of $70 \times 45 \times 45$ cm ($L \times B \times H$), filled with 25 cm of aged tap water with a filtration system, and kept in a acclimatized animalarium with a 12/12-h day-and-night cycle. To collect skin secretion, each frog was placed in a plastic ziplock bag and manually massaged to simulate the swallowing motion and esophageal peristalsis of a predator (as expected during an attack). The secretion produced during this stimulation tends to stick to the inner surface of the bag and is therefore easy to collect. After 5 min, all secretion on the frog and the bag's inner surface was collected in a pre-weighed 1.5 ml microcentrifuge tube using a spatula, briefly centrifuged at 10,000 r.p.m. and weighed again to obtain a net weight for the skin secretion. Obtained skin secretion samples ranged between 85.7 and 419.8 mg, roughly corresponding to 85–420 μl. The secretion samples were subsequently diluted to obtain 10 ml of a 9 M urea/0.1% TFA solution, and fully dissolved by brief vortexing. A 100 μl sample of each solution was used to measure caerulein amounts and concentrations via the ELISA procedure described above. Using the ELISA results, we inferred that the individual frogs produced between 0.12 and 4.27 mg caerulein, amounts that are high, but comparable to those estimated for several peptide toxins in other frogs[10,39,40]. Combined with the estimated skin secretion volumes, the estimated amounts correspond to skin secretion concentrations between 1.0 and 7.5 mM, and the lowest limit of this range was used as a biologically realistic concentration for in vivo experiments (see below).

**In vivo quantification of absorbed radiolabeled caerulein.** To investigate toxin absorption in a live model predator, radiolabeled caerulein was quantified in the blood and organs of snakes of the species Thamnophis eques (Colubridae: Natricinae) after oral administration in the presence vs. absence of CPF. Adult female snakes of this species ($n = 18$) were purchased from J & M Garters (Spijkenisse, The Netherlands), and housed in $70 \times 40 \times 20$ cm plastic boxes ($L \times B \times H$), kept in a acclimatized animalarium with a 12/12-h day-and-night cycle at 20–25 °C. A first experiment was designed to test a biologically realistic peptide concentration, and involved a comparison of two treatments ($n = 3$): (1) oral administration of 50 μl buffer solution, containing 50 nmol caerulein + 50 nmol CPF (i.e., both at 1 mM); and (2) oral administration of 50 μl buffer solution containing 50 nmol (1 mM) caerulein alone. A sample size of three was chosen because it allows for sufficient statistical power using linear mixed modeling (see below) at a minimum cost of animal lives. A second experiment was designed to test a peptide concentration corresponding to our in vitro tests, which involved the following two treatments ($n = 6$): (1) oral administration of 50 μl buffer solution containing 5 nmol caerulein + 5 nmol CPF (i.e., both at 100 μM); and (2) oral administration of 50 μl buffer solution containing 5 nmol (100 μM) caerulein alone. Here a sample size of six was chosen because we suspected a lower effect size than in the previous experiment due to the lower peptide concentrations.

For each experiment, snakes were selected for both treatments to obtain identical distributions of BWs. Caerulein-DTPA (65 or 375 nmol) was labeled with 111InCl3 (65–90 MBq) (Mallinckrodt Pharmaceuticals) in 0.2 M NH4OAc (pH 5.0; ammonium acetate) by incubation at 50 °C for 30 min (total volume 413 μl). Radiochemical purity (>95%) was verified by iTLC on silica gel (Pall Corporation) using 0.1 M sodium citrate buffer (pH 5.0) as mobile phase. Snakes were anesthetized with an intracardial injection of propofol (12 μg/g BW) and subsequently positioned on their back. Peptide solutions were orally administered with standard syringes by carefully dripping the content (50 μl of a 1 mM or 100

µM buffer solution) on the palate of the snakes, after which the remaining radioactivity of the syringe was measured to determine the radioactivity of actual administered oral dose. Blood samples (10–50 µl) were taken from the heart with standard insulin syringes at predetermined time points after the start of the experiment (defined as the moment of oral peptide administration). After 30 min, snakes were euthanized with an intracardial injection of T61 (0.5 µg/g BW) and organs and tissues were collected and weighed. Organ and tissue radioactivity was measured in a gamma counter against a standard of known activity and corrected for the decay in radioactivity. The investigator conducting these measurements was blinded to the oral administration treatments (presence vs. absence of CPF). Before the onset of the experiment, it was decided that samples would be excluded from the analysis only if radioactive contamination during the dissection was deemed highly probable. This was the case for one measurement for the brain (one snake treated with 1 mM caerulein + CPF), where radioactive contamination from the oral cavity during excision of the brain from the skull could not be ruled out. Caerulein levels in the snakes' blood and organs were expressed in pmol/gram (blood or tissue). Unlike experiments based on intravenous injection, it is impossible to infer absolute absorption efficiencies based on the total amount of toxin in a subject's body at any time after oral administration. Yet, to allow comparison with doses based on whole BWs, as reported in previous studies on caerulein toxicity, the obtained values were corrected for the proportion of blood in a snake's body (5–8%)[29], yielding values in pmol/gram BW. Note that these values may represent an increasingly larger underestimation of the actual amount of toxin absorbed by a snake's body, since an increasingly large fraction of the orally administered dose will have been transferred from the blood to other tissues. As an additional measure for systemic toxin exposure, we calculated for each snake the AUC value (expressed in pmol×min/g). The AUC value incorporates multiple successive measurements of the amount of toxin in the blood, and is calculated as the area under the connecting lines between successive data points as defined by the commonly used "trapezoidal rule" method[54–56].

**Statistics**. All data series were tested for normality and equality of variance and subsequently analyzed using SPSS version 23.0 (IBM Corp, USA; most tests) and R version 3.1.1 (https://www.r-project.org; linear mixed models). Differences in TEER among peptide treatments were evaluated using one-way analysis of variance with a Tukey (equal variances assumed) or Games–Howell (equal variances not assumed) post hoc test. Pairwise differences in transepithelial toxin passage between treatments (ELISA) were evaluated using Student's t-tests. Linear mixed models considering "time" and "treatment" (different peptide administrations) as fixed factors and "snake individual" as random factor were used to test for differences in caerulein blood levels over time between treatments. Linear mixed models considering "organ" and "treatment" as fixed factors and "snake individual" as random factor were used to evaluate their effects on caerulein levels in organs and blood after 30 min.

**Data availability**. The data that support the findings of this study are available from the corresponding author upon reasonable request.

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

## Acknowledgements

We thank Jannie Smit for kindly providing the photograph in Fig. 1a, Marjolein Couvreur for assistance with the scanning electron microscopy, Bram Vanschoenwinkel for help with statistics, and Franky Bossuyt, Keely Smith, Sunita Janssenswillen, and Ines Van Bocxlaer for commenting on earlier versions of the manuscript. This work is financed by grant no. G0D3214N of FWO-Vlaanderen. K.R., V.C. and S.B. received additional support from SRP-groeiers grants SRP-30 of Vrije Universiteit Brussel.

## Author contributions

A.M., F.P., K.R., S.B. and V.C. conceived the project and acquired funding; A.M., C.R., F.P., K.R., S.H., T.H. and V.C. designed the research; C.B. and S.B. designed and synthesized peptides; C.R. and E.V. conducted the in vitro experiments; C.R., M.M. and W.B. prepared the scanning electron microscopy; A.M., C.P., C.R., F.P., K.R., S.H. and T.H. conducted the in vivo experiments; C.R., S.H. and K.R. analyzed the data; all authors contributed to the manuscript.

## Additional information

**Competing interests:** The authors declare no competing financial interests.

