## [Peer Review File · Nature Communications]

Reviewers' comments:

Reviewer #1 - Expertise: AMP mode of action, membrane permeability of peptides

This paper presents the intriguing hypothesis that some "antimicrobial" peptides from frog skin secretions may actually comprise a system for delivery of co-secreted toxins through the oral/gut epithelia, rather than (or in addition to) their long suspected role as elements of innate immunity. The authors show in vitro experiments in cell culture, as well as in vivo experiments to support their hypothesis. If this is true, it could possibly have a very significant impact on drug delivery approaches for peptide and protein drugs.

While the hypothesis is intriguing, and the data generally seem to support it, I feel that some additional information is needed to judge the likelihood that the reported effect is biologically important.

Major comments:

1) In figure 1b the authors only show data for CPF and Cae separately, while in Figure 1c (TEER) they show CPF only in mixtures with Cae, and not separately. Is there a reason for this? For completeness, there should be data for TEER with AMP alone and LDH release with AMP+toxin.

2) In figure 1e and in the methods section, the starting concentration of toxin and AMP are shown in micromolar, but the measured basolateral concentration is shown in ng/ml. Why were different units used for the same molecule in one plot? The trans concentration must be shown in the same units as the starting concentration. The same thing applies to Figure 3d.

3) Figure 3d, bottom. Are the units ng/ml or ug/ml? The authors should use the same units as the other three plots of this experiment.

4) If I convert the basolateral toxin concentration in Figure 1e to uM, assuming 1.5 kDa MW, I get 0.8 uM for Caco-2 cells and 0.1 uM for the Oral monolayer model, or 0.8% and 0.1% respectively. The authors should make a convincing argument that this seemingly very low efficiency of transport could be biologically relevant.

5) In the whole animal experiments, Figure 1c reports delivery as "%dose/g" while the text describes area under the curve for %dose per minute per gram. What are the units in Figure 1b and 1c? For non-specialists, there should be a better description of how this value is calculated and what it means. The authors should probably convert this data to standard units of concentration, at least for the peak concentration. Is this peak concentration enough to have a biological effect (see next comment)?

6) Because the starting AMP and toxin concentrations in the whole animal experiments are similar to those measured in frog secretions, the authors must show that the absorbed caerulein concentrations are sufficient to have a dramatic biological effect on the predator.

7) The concentration of caerulein delivered to tissues in animals is measured by radioactivity. The authors must show that the caerulein in tissues is intact toxin. One can readily imagine that AMP-lysed cells will release proteases that degrade the toxin immediately, releasing the radioisotope which may then follow other means of transport. In addition, plasma proteases/peptidases may rapidly degrade the toxin during transport. Biological half-lives of some peptides are as short as 1-5 minutes. Others can persist for hours.

8) What is the evidence that the toxin, caerulein needs to obtain a systemic distribution to have a

biological effect? There are receptors in the gut, for example. Similarly, do dermorphin or bombesin have toxic effects in the gut? Kin selection would support a toxic mechanism that occurs even after ingestion of the prey item.

Minor Comments

- 1) It would be very helpful to have Supplementary Table 1 in the main paper, if possible.
- 2) Please change "100uM" to "100 μ M"

Reviewer #2 - Expertise: Structures of animal toxins

This study by Raaymakers et al. provides experimental evidence for the hypothesis that frog skin antimicrobial peptides serve an important role in enhancing the efficacy of co-secreted toxins. Most experimental data is obtained for the combinatorial effect of Cae and CPF but the study also expands to two other co-secreted AMP-toxin systems in order to provide additional support for the hypothesis of co-action.

As stated by the authors the implications of these findings are important, provide insight into our understanding of frog defensive and challenge current concepts on what is considered a poisonous versus a venomous animal. It's a beautiful showcase of the evolutionary cleverness of poisonous animals and will be well received in the community and wider NatCom readership.

The evidence is compelling and provides a good balance between in vivo proof-of-concept studies without unnecessarily sacrificing too many animals. The experiments are well designed and conducted and I only have a few minor suggestions.

Specific comments:

p1 L33: replace taxa with animal or alternative wording, mechanisms are not always taxa-specific

p2 L56: should read "than" not "that"

Fig 1:

It would have been good to evaluate the effect of AMP alone on TEER without addition of toxins (i.e. CPF, BLP and Drs).

What snake is shown in the image and does this frog species (*Litoria caerulea*) also co-secrete toxins and AMPs? I assume that no picture was available for the frog species studied here.

p3 L92: Only small toxins were tested and the authors may want to be more careful when generalizing this finding to other toxins that may potentially be larger and require active uptake into the cell.

Is there any behavioral toxicity data available for the snakes used as predators? Did they show any signs of intoxication prior to dissection?

Fig 3C: What was the effect of BLP and Drs on TEER alone (see comment for Figure 1)?

Choice of toxins and antimicrobial peptides: Would the experiments have worked if the peptides and co-secreted AMPs had not been taken from the same species? Is there a co-evolution between these two systems? I do not think that these experiments are necessary for the current manuscript but it would be interesting to investigate this in the future.

Could the authors comment on other predators of frogs and how their predation behavior is similar or different to snakes (e.g. do predators generally eat/bite their frog prey or do some snakes strangulate their frog prey first in which case the toxin delivery system must work on skin rather than the epithelium of the oral cavity. Is it known whether predators have evolved behaviors to avoid being exposed to the secretion system (e.g. target the belly rather than back of the frog)?

Material and Methods:

Peptide synthesis: provide details on peptide cleavage from resin, purification and quantification

In vivo snake assays:

How was the toxin/AMP administered orally? Through a syringe etc.?

Reviewer #3 - Expertise: Novel modes of venom delivery

The manuscript proposes a novel ecological role of frog defensive secretion AMPs as spreading factors that enable the actions of other defensive peptide toxins from the same secretions. The results in many ways resolves the conundrum of how bioactive peptides from frog skin secretions and other 'poisons' are able to act as defensive agents given their inability to effectively cross oral epithelia – a property usually considered crucial to toxic components of poisons due to the lack of physical delivery structures. The results also provide an exciting example of synergy in bioactive animal secretions. Even more interesting, in my opinion, is that this specialised chemical 'delivery' mechanism highlights the grey area that exists between the distinctions between poisons (passively delivered toxins) and venoms (actively delivered toxins).

In summary, the manuscript casts new light on the evolution of frog defensive skin secretions, and will hopefully inspire similar research into the function and evolution of peptide-toxin-containing bioactive secretions from other animal groups. I think the paper challenges the existing dogmas of toxinology and that it will no doubt be perceived as controversial by many. This expected controversy highlights the impact of the paper rather than the lack of well-founded conclusions, which I think could be even further improved by addressing the two questions below:

The first and most obvious question is whether the time it takes for caerulein to act is short enough to be significant to prevent being killed by a predator. As the authors point out in the first sentence of the introduction, the prey only has minutes to induce a response in the predator to avoid being killed. It is therefore not the effect of CPF on the amount of caerulein reaching tissues that is interesting, but its effect on the time it takes for caerulein to produce an effect. Although the co-administration of CPF and caerulein resulted in a significant increase in blood and tissue levels after 30 minutes, it would be interesting to know how long it would take for caerulein to reach local tissue abundances that based on its potency might be enough to produce an aversion response in the predator, with and without co-administration with CPF.

Along the same lines, the poreforming activity of AMP's would presumably be painful to predators as well, and possibly induce a faster pain response than caerulein? Is there evidence that these AMPs do, or do not, also act as defensive peptides themselves?

In addition I have a few specific, very minor comments

I think it would be helpful to the reader to point out the effect that a potent cholecystokinin receptor agonist such as caerulein would have in a predator, not just where cholecystokinin receptors are expressed (line 104).

It would be good to mention examples where spreading factors have been shown in other animals that use chemical defense mechanisms without physical generation of a wound. Whipscorpions such as *Mastigoproctus*, for example, contain caprylic acid to facilitate penetration of their (mostly acetic acid based) defensive secretion (Eisner et al. 1961 *Journal of Insect Physiology*, 6 (4):272-298. [https://doi.org/10.1016/0022-1910\(61\)90054-3](https://doi.org/10.1016/0022-1910(61)90054-3)).

The first listed implication of the results identified by the authors is a novel potential therapeutic application of frog AMPs. However, if the mode of action of these AMPs in reducing membrane integrity is poreformation, how could cytotoxicity be reduced while the activity that is the base of the potential application be preserved?

Replace comma after '(presence vs. absence of CPF),' on line 364

Please include time in the legends of figures 1 b, d, e, 2c, and 3 b, d.

Reviewer #4 - Expertise: Poisonous amphibian biology & ecology

This manuscript describes a molecular toxin-delivery system in three frog species. It is well known that many amphibians have bioactive peptides in skin secretions, but this paper provides evidence that co-expressed proteins act as wound-inducing agents to allow systemic entry of peptide toxins. The authors overall did a great job pursuing this question from many angles, including in vitro and in vivo (snake predator) assays. A second major strength is the evolutionary perspective the study brings by examining this phenomenon in many frog species.

I have two remaining questions after reading this paper:

First, the evidence that toxic peptides evolved three independent times is weak and is likely due to the poor breadth of sequencing among amphibians. I commend the authors for taking the effort to study three species - this shows their observed phenomenon may be a general biological principle rather than an odd observation in an odd frog. However, the evolutionary origins of peptides is far from understood and care should be taken in how the authors claim "convergent evolution". A minor note here is that the authors claim in the title this observation applies to amphibians, but as this research group knows well, there are other amphibians besides anurans and care should be taken in how broadly they apply this claim.

My second concern is about the timing of peptide effect and what this means for predator-prey interactions. When a snake eats a frog, is there even time for this peptide effect in the oral cavity to have any effect? Is five minutes plenty of time or no time at all? The timing should be better explained in the context of fitness in the discussion.

Reviewer #1

1) *In figure 1b the authors only show data for CPF and Cae separately, while in Figure 1c (TEER) they show CPF only in mixtures with Cae, and not separately. Is there a reason for this? For completeness, there should be data for TEER with AMP alone and LDH release with AMP+toxin.*

Similar comments were made by Reviewer #2 (remarks #3 and #6). We therefore performed extra LDH experiments of the AMP + toxin pairs and extra TEER experiments of the AMPs alone. These are now incorporated in the *Results* section (p. 3, yellow for Cae + CPF; p. 5, yellow for Bom + BLP and Drm + Drs) and the appropriate figures (Fig. 1c, Supplementary Fig. 1). To maintain consistency with the illustrated TEER and ELISA data, LDH results based on the AMPs alone (Figs. 1b and 3b in the original manuscript) have now been replaced by results based on the two peptides combined (AMP + toxin; Figs 1c and 3c in the revised manuscript). The results based on the AMPs alone have been moved to two additional supplementary figures (Supplementary Figs. 1 and 5). For completeness, we also added a SEM photograph to Supplementary Fig. 1 visualising the effect of the AMP alone on Caco-2 epithelia.

In short, these extra analyses reveal a near-identical effect on epithelia of administering the toxin + AMP or the AMP alone, confirming that the AMP is the essential component in frog skin secretion for toxin absorption.

2) *In figure 1e and in the methods section, the starting concentration of toxin and AMP are shown in micromolar, but the measured basolateral concentration is shown in ng/ml. Why were different units used for the same molecule in one plot? The trans concentration must be shown in the same units as the starting concentration. The same thing applies to Figure 3d.*

We agree that using the same units would allow for better comparison of the amounts of toxin at the apical and basolateral sides. Since expressing both sides in the same unit of concentration (either micromolar or ng/ml) would still yield incomparable values (due to the different volumes of the apical insert and basolateral well), we now express the caerulein quantity at both sides in nanomol (nmol). Figures 1e and 3d have been modified accordingly.

3) *Figure 3d, bottom. Are the units ng/ml or ug/ml? The authors should use the same units as the other three plots of this experiment.*

As requested, the basolateral concentration in Fig. 3d (now Fig. 3e; see remark #9 why) has been changed to the same unit (nmol) as all other ELISA experiment figures. We changed from ng / ml to nmol to allow better comparison with the applied dose (now also mentioned in the text in μM and nmol; see remark #2 above).

4) *If I convert the basolateral toxin concentration in Figure 1e to μM , assuming 1.5 kDa MW, I get 0.8 μM for Caco-2 cells and 0.1 μM for the Oral monolayer model, or 0.8% and 0.1% respectively. The authors should make a convincing argument that this seemingly very low efficiency of transport could be biologically relevant.*

By expressing both apical and basolateral peptide amounts in the same unit (nmol), the absorption efficiencies are now easier to infer from our ELISA figures and are also mentioned in the Results text (p. 3, green). Although still low, they are higher than calculated by the reviewer due to the fact that apical and basolateral volumes differed in our setup (apical insert = 200 μL , basolateral well = 1000 μL ; we think that the reviewer assumed equal volumes). With the correct volumes and MW of caerulein, we calculate that 0.3 – 1 % of caerulein is transferred in the absence of the AMP, and 2.3 –

6.8 % in its presence. In the Results section, we argue that although these values are still fairly low, they show a major effect of AMP co-secretion, which is what this experiment was designed for (p. 3, blue).

We interpreted this point, together with the following two remarks (#5 and #6), as a request for a better argumentation of the biological relevance of the absorption enhancement as demonstrated here. We therefore added a sentence to the Discussion section (p. 6, blue) arguing that the *in vitro* observed absorption efficiency is sufficiently high in light of the high toxin concentrations typically found in a frog's skin (reported or measured in this study).

5) *In the whole animal experiments, Figure 1c reports delivery as “%dose/g” while the text describes area under the curve for %dose per minute per gram. What are the units in Figure 1b and 1c? For non-specialists, there should be a better description of how this value is calculated and what it means. The authors should probably convert this data to standard units of concentration, at least for the peak concentration. Is this peak concentration enough to have a biological effect (see next comment)?*

Reviewer #1 is probably referring at Fig. 2b and 2c (*in vivo* results) instead of Fig 1 (*in vitro* results). To clarify the biological meaning of the units used in the figures showing our *in vivo* results, a detailed description of the units (Figure 2 and Supplementary figure 3), why they were chosen, and how they are calculated is provided in the Methods section (p. 14, yellow). To allow comparison with previously published concentrations, we now express the caerulein blood levels in pmol/ g blood in Fig. 2 and the Results section and also convert them to values expressed in pmol/ g body weight (BW), taking into account that a snake's blood takes 5–8 % of its body (p.4, pink).

As far as we know, the threshold dose of caerulein or any other frog toxin required to cause adverse effects in a snake (or any other natural predator of amphibians) has never been inferred. Obtaining these values would require *in vivo* experiments with unsedated snakes that would exceed conventional ethics and would be very hard to obtain approval for by an ethical committee. However, because this is an important aspect, we compared the concentrations to those reported in early (1960s and 1970s) toxinological studies on various amniotes (dogs, cats, rats and chicken; see Erspamer V. (1970) Progress Report: Caerulein. *Gut* 11: 79–87). We now mention in the Results section (p. 4, blue) that although no toxinological data are available for caerulein in snakes, the observed concentrations after ten minutes reach levels that exceed those reported to cause adverse effects in other vertebrates (1 µg / kg BW, corresponding to 0.74 pmol/ g BW, injected intravenously).

6) *Because the starting AMP and toxin concentrations in the whole animal experiments are similar to those measured in frog secretions, the authors must show that the absorbed caerulein concentrations are sufficient to have a dramatic biological effect on the predator.*

As mentioned above (see remarks #4 and #5), we added several sentences to the Results and Discussion sections (blue section on p. 3, blue, p. 4, and p. 6) to make a better case for the biological relevance of the observed absorption enhancement.

7) *The concentration of caerulein delivered to tissues in animals is measured by radioactivity. The authors must show that the caerulein in tissues is intact toxin. One can readily imagine that AMP-lysed cells will release proteases that degrade the toxin immediately, releasing the radioisotope which may then follow other means of transport. In addition, plasma proteases/peptidases may rapidly degrade the toxin during transport. Biological half-lives of some peptides are as short as 1-5 minutes. Others can persist for hours.*

Although peptide degradation is an important issue, it is unlikely that the contribution of AMPs to caerulein degradation by releasing intracellular peptidases would invalidate our results. First, our ELISA experiments show higher levels of transferred caerulein in the presence of CPF vs. in its absence, which, in the case of AMP-aided degradation would be improbable. Second, degradation by

released intracellular peptidases would be a critical problem faced by *any* frog co-secreting AMPs and toxins. The argument of plasma peptidases is a very valid one (although also faced by all frogs secreting peptide toxins with systemic targets). To anticipate similar concerns of future readers, we now mention in the Introduction that post-translational modifications are a general property to extend half-lives of amphibian peptide toxins in a predator's bloodstream (p. 2, grey), and in the Results section, published long half-lives in blood plasma and tissue are now mentioned specifically for the three model toxins under investigation (grey sections on p. 3 and 5). We also state in the Results section how radiolabeling, despite potential degradation of the peptide, is a useful way to compare absorption efficiencies under different treatments (p. 4, grey).

8) *What is the evidence that the toxin, caerulein needs to obtain a systemic distribution to have a biological effect? There are receptors in the gut, for example. Similarly, do dermorphin or bombesin have toxic effects in the gut? Kin selection would support a toxic mechanism that occurs even after ingestion of the prey item.*

We added sentences in the Results section to better introduce the three toxins considered in this study and explain that they *all* require epithelial absorption to induce their toxic effects (p. 2-3, pink, and p 5, pink). For caerulein, we explain that cholecystinin receptors (CCKR), the target receptors of caerulein, are only accessible through absorption, even those in the gut. CCKR in the gut occurs on the basolateral side (within the tissue), not on the apical side (the gut's lumen). Similarly, we explain that dermorphin binds opioid receptors of nerve cells and bombesin binds bombesin/neuromedin receptors on smooth muscle cells surrounding veins and the gut.

If a frog would be ingested and eventually passed on to the small intestine, its toxins would still need to cross intestinal epithelia to intoxicate the predator. If the frog dies in the process, such scenario could indeed be beneficial to the remaining frog population, which we believe is what Reviewer #1 is referring at. To address this possibility, we now postulate in the Discussion section (p. 7, yellow) that if the frog is killed, its toxic effects on the predator could still be beneficial to the remaining population, potentially aided by a predator's learning response. However, we prefer to avoid the term "kin selection" since this term is typically used in a very specific microevolutionary context where a selective benefit applies only to closely related individuals *within* a population. In the case of toxic frogs, the benefit may apply equally to any member of the remaining population, regardless of its relatedness of the dying frog.

9) *It would be very helpful to have Supplementary Table 1 in the main paper, if possible.*

We have now integrated the peptide sequences listed in Supplementary Table 1 in Fig. 1b and Fig. 3b. This causes a shift in the figure numbers with respect to the previous manuscript version (e.g. Fig. 1c in the original manuscript is now Fig. 1d)

10) *Please change "100uM" to "100 μM"*

This typo has been corrected.

Reviewer #2

1) *Replace taxa with animal or alternative wording, mechanisms are not always taxa-specific.*

The suggested replacements have been made.

2) *p2 L56: should read "than" not "that"*

The suggested replacement has been made.

3a) *It would have been good to evaluate the effect of AMP alone on TEER without addition of toxins (i.e. CPF, BLP and Drs).*

This remark is related to remark #1 of Reviewer #1. We performed extra LDH and TEER experiments of the three tested AMPs, and incorporated the results in the *Results* section (yellow sections on p. 3 and p. 5) and the appropriate figures. To maintain consistency, LDH results based on the AMPs alone in the paper figures (original Fig. 1b and 3b) are now replaced by results based on the two peptides combined (AMP + toxin) (Figs 1c and 3c in the revised manuscript). The results based on the AMPs alone have been moved to two additional supplementary figures (Supplementary Figs. 1 for CPF and supplementary Fig. 5 for bombesin and dermaseptin). As indicated above, our extra experiments reflect those of AMP + toxin co-administration and confirm that the AMP and not the toxin itself stands in for epithelial permeation.

3b) *What snake is shown in the image and does this frog species (Litoria caerulea) also co-secrete toxins and AMPs? I assume that no picture was available for the frog species studied here.*

Unfortunately, we do not have a picture of a snake preying on any of our model frog species. The present picture illustrates the precarious first moments of a poisonous frog under attack by a predator and served as a major inspiration for our study (the frog survived this attack by the way). However, the *Litoria caerulea* frog shown in the picture does secrete caerulein as its main defensive toxin (like *Xenopus* frogs), alongside a rich diversity of broad-spectrum antimicrobial peptides (related to dermaseptins). The snake is the Australian tree snake *Dendrelaphis punctulatus* (Colubridae). To increase the relevance of the picture, the names of both the frog and the snake are now mentioned in the figure caption and we also mention that the frog secretes caerulein, similar to *X. laevis*.

4) p3 L92: *Only small toxins were tested and the authors may want to be more careful when generalizing this finding to other toxins that may potentially be larger and require active uptake into the cell.*

This is a valid point, and the only reason why we did not test larger toxins (like anntoxin or prokineticin-related toxins) is the difficulty to synthesize those toxins in sufficient amounts (possibly requiring recombinant techniques instead of solid-phase synthesis). To relativise our results in light of large toxins, we added a segment to the Discussion to argue that although the three chosen peptides span the typical size range of the majority of frog toxins, the question remains whether larger toxins would equally benefit from absorption enhancement (p. 6, green), and that our ELISA results indeed suggest the existence of a size dependency.

5) *Is there any behavioral toxicity data available for the snakes used as predators? Did they show any signs of intoxication prior to dissection?*

As far as we know, there are no behavioral data related to intoxication for the snake species we worked with. Behavioral data exist for a related species (*Thamnophis sirtalis*) after eating toxic newts and suffering the effects of the neurotoxin tetrodotoxin (TTX), but since this toxin is readily absorbed (and not a peptide), it is hardly relevant to our study. Unfortunately, since our snakes were under anesthesia during the whole experimental procedure (a key directive of our ethical permits for this *in vivo* experiment), behavioral data could not be monitored. Basically, our *in vivo* tests were specifically designed to measure the difference of toxin absorption in the presence vs. absence of a cosecreted AMP.

6) *Fig 3C: What was the effect of BLP and Drs on TEER alone (see comment for Figure 1)?*

This remark is related to remark #1 of Reviewer #1 and remark #3 of reviewer #2. As explained above, we performed extra LDH and TEER experiments of the three tested AMPs, and incorporated the data in the *Results* section (yellow sections on p. 3 and 5). The results of the AMP + toxin are now shown in Fig. 1 and 3, while the results for the AMPs alone are shown in the supplementary Figs. 1 and 5. The new results confirm the activity of the AMP on epithelia, confirming its contribution to toxin absorption.

7) *Choice of toxins and antimicrobial peptides: Would the experiments have worked if the peptides and co-secreted AMPs had not been taken from the same species? Is there a co-evolution between these two systems? I do not think that these experiments are necessary for the current manuscript but it would be interesting to investigate this in the future.*

We agree with the reviewer that this is an interesting question. However, given the relatively crude mechanism by which AMPs increase the permeability of the epithelia (by damaging cell membranes and causing ruptures in the cell layer, see Fig. 1e), we find it unlikely that these AMPs act specifically for cosecreted toxins, or co-evolved with them.

8) *Could the authors comment on other predators of frogs and how their predation behavior is similar or different to snakes (e.g. do predators generally eat/bite their frog prey or do some snakes strangulate their frog prey first in which case the toxin delivery system must work on skin rather than the epithelium of the oral cavity. Is it known whether predators have evolved behaviors to avoid being exposed to the secretion system (e.g. target the belly rather than back of the frog)?*

There are various predators known to have either evolved resistance to the effect of prey toxins or learned to avoid the toxic skin of a frog. Although this is relevant as an adaptation to toxic prey in general, it is not more so as an adaptation to absorption enhancement, making it perhaps less relevant to specifically discuss this in the present manuscript. However, we agree with the reviewer that some predatory behaviors that kill or severely injure their prey before eating it, would render the toxin system incapable to increase the prey's survival chance, leaving only the possibility of an effect through kin selection. As mentioned above (see remark #8 of reviewer #1), we added a section about toxin absorption and its possible effectiveness against predators with different prey handling capacities in the Discussion (p. 7, yellow).

9) *Material and Methods: Peptide synthesis: provide details on peptide cleavage from resin, purification and quantification.*

More details concerning the peptide synthesis have been added in the Methods section (p. 8-9, yellow).

10) *In vivo snake assays: How was the toxin/AMP administered orally? Through a syringe etc.?*

The peptide solutions (50 µl drops) were carefully applied on the palate of anesthetized snakes using standard insulin syringes. The solution was dripped on the palate, not injected. Radioactivity of the syringes was measured before and after peptide administration to determine the exact oral starting dose of radiolabeled caerulein that individual snakes received. The exact procedure of peptide administration and snake handling is now explained in more detail in the Methods section (p. 13, green).

Reviewer #3

1) *The first and most obvious question is whether the time it takes for caerulein to act is short enough to be significant to prevent being killed by a predator. As the authors point out in the first sentence of the introduction, the prey only has minutes to induce a response in the predator to avoid being killed. It is therefore not the effect of CPF on the amount of caerulein reaching tissues that is interesting, but its effect on the time it takes for caerulein to produce an effect. Although the co-administration of CPF and caerulein resulted in a significant increase in blood and tissue levels after 30 minutes, it would be interesting to know how long it would take for caerulein to reach local tissue abundances that based on its potency might be enough to produce an aversion response in the predator, with and without co-administration with CPF.*

We agree with the reviewer, but the amount of toxin reaching tissues after a specific time, and the rate at which effective toxin levels are reached, are correlated. As pointed out above (see remark #5 of reviewer #1), the caerulein level required to cause adverse effects are unknown for snakes and near-impossible to assess in an ethically acceptable manner. But from the plots in our Fig. 2b, it is obvious that threshold levels of caerulein in the blood, whatever they may be, will be reached earlier in the presence of CPF, than in its absence. Caerulein levels in the blood cannot be measured continuously throughout the experiment to record the time when such level would be reached, and need to be based on blood samples taken at prespecified time points. Organ levels can only be sampled after euthanising the snakes. As a result, feasible and reliable sampling only allows a basic estimation of absorption time, e.g. by inferring the slope between successive sampling points in Fig. 2b.

Besides that, we admit that our first sentence of the introduction (cited by the reviewer above) may have been slightly exaggerated in light of natural history reports on how long the fight between frogs and their predators may last.

To address the reviewer's concern, we did the following:

- we toned down the first sentence of the introduction, now mentioning "little time" instead "only minutes" (p. 1, green).
- We specifically emphasize in the Results section that the chosen 30-minutes time window is biologically realistic in light of previous natural history reports (p. 4, green). The same reports are cited again in the Discussion section, where we briefly mention that the fight between frogs and predators may take up to 50 minutes (p. 7, green).
- As mentioned above, we specifically mention in the Results section that the caerulein levels in the snakes' blood at 10 and 30 minutes after oral administration exceed those that reportedly cause adverse effects in other vertebrates (p. 4, blue).

2) *Along the same lines, the pore-forming activity of AMP's would presumably be painful to predators as well, and possibly induce a faster pain response than caerulein? Is there evidence that these AMPs do, or do not, also act as defensive peptides themselves?*

We agree with the reviewer that pain/irritation could be an additional mode of action of AMPs and since there is currently no evidence for this hypothesis, we are currently investigating this potential additional function. The primary objective of the present manuscript however, is to provide an explanation for how frogs manage to get their peptide toxins into a predator's bloodstream. To acknowledge the possibility that AMPs additionally act as analgesics/irritants to oral mucosa during the attack, we added a sentence to the Discussion that postulates this hypothesis (p. 6, pink).

3) *I think it would be helpful to the reader to point out the effect that a potent cholecystokinin receptor agonist such as caerulein would have in a predator, not just where cholecystokinin receptors are expressed (line 104).*

As suggested, we now added information on the toxic effects of caerulein in the Results section, right after we discussed its target CCK receptors (p. 2, pink).

4) *It would be good to mention examples where spreading factors have been shown in other animals that use chemical defense mechanisms without physical generation of a wound. Whipscorpions such as Mastigoproctus, for example, contain caprylic acid to facilitate penetration of their (mostly acetic acid based) defensive secretion (Eisner et al. 1961 Journal of Insect Physiology, 6 (4):272-298. [https://doi.org/10.1016/0022-1910\(61\)90054-3](https://doi.org/10.1016/0022-1910(61)90054-3)).*

We like to thank the reviewer for pointing out this example, and integrated it into the Discussion section (p. 7, blue).

5) *The first listed implication of the results identified by the authors is a novel potential therapeutic application of frog AMPs. However, if the mode of action of these AMPs in reducing membrane integrity is poreformation, how could cytotoxicity be reduced while the activity that is the base of the potential application be preserved?*

Actually, the first implication of our results was more meant as a cautionary tale: AMPs, currently investigated as potential new antibiotics, are cytotoxic to vertebrate epithelia and this should be taken into account by pharmacologists. Because this is a rather pessimistic message affecting a large research field, we preferred to end on a more positive note, mentioning a potential alternative research avenue related to oral drug delivery. This note was inspired by the fact that permeation enhancers (including toxins of other organisms) are currently under investigation as a potential strategy to overcome the problem of oral peptide delivery. Similar to frog AMPs, such permeation enhancers face the same problem of toxicity and typically peptidomimetic modifications are explored to increase their therapeutic window. However, we have no clear-cut answer to the reviewer's question how frog AMPs could be optimized to enhance oral drug delivery. We therefore decided that it would be safer to remove this sentence from our discussion.

6) *Replace comma after '(presence vs. absence of CPF),' on line 364*

The suggested adaptation has been made.

7) *Please include time in the legends of figures 1 b, d, e, 2c, and 3 b, d.*

Time intervals of experiments have been included in all figure legends where appropriate.

Reviewer #4

1) *First, the evidence that toxic peptides evolved three independent times is weak and is likely due to the poor breadth of sequencing among amphibians. I commend the authors for taking the effort to study three species - this shows their observed phenomenon may be a general biological principal rather than an odd observation in an odd frog. However, the evolutionary origins of peptides is far from understood and care should be taken in how the authors claim "convergent evolution". A minor note here is that the authors claim in the title this observation applies to amphibians, but as this research group knows well, there are other amphibians besides anurans and care should be taken in how broadly they apply this claim.*

The convergent evolution of AMPs in three independent frog lineages was postulated by König et al (2011), admittedly based on circumstantial evidence. However, a more recent study (Roelants et al.

PLoS Genetics, 2013) provided phylogenomic evidence for the origin of AMPs in the family Pipidae, to which *X. laevis* - one of the model species investigated here - belongs. This evidence implies that pipid AMPs (including CPF) evolved independently from the AMPs of other frog families. We could elaborate on this evidence to make this argument more solid in our manuscript but it would divert too much from the main message of our study. Instead, we opted to follow the reviewer's advice and be more careful in the statement of convergent evolution. We now place more emphasis on their widespread distribution rather than their independent origins. Similarly, based on the reviewer's side remark, we replaced "amphibians" by "frogs" in the manuscript title and throughout the text, where appropriate.

2) My second concern is about the timing of peptide effect and what this means for predator-prey interactions. When a snake eats a frog, is there even time for this peptide effect in the oral cavity to have any effect? Is five minutes plenty of time or no time at all? The timing should be better explained in the context of fitness in the discussion.

This remark is similar to remark #1 of reviewer #3. To address both remarks, we added sentences to Results and Discussion citing previous natural history reports, indicating that the chosen time frame of 30 minutes is biologically relevant (p. 4, green) as the fight between frogs and predators may take up to 50 minutes (p. 7 green).

REVIEWERS' COMMENTS:

Reviewer #1 (Remarks to the Author):

The hypothesis tested in this paper remains fascinating, well supported by this work, and potentially very important. Here, the authors make an even more compelling case for it. In this revised version the authors have added new discussions, much new data and a much better discussion. While there are questions that remain, I feel that this manuscript is ready for publication, and the remaining questions should be addressed by the larger scientific community.

Reviewer #2 (Remarks to the Author):

The authors have satisfactorily addressed my comments and I have no more suggestions.

Reviewer #3 (Remarks to the Author):

All my concerns have been addressed by the authors, and I look forward to seeing the manuscript in its published form.

Reviewer #4 (Remarks to the Author):

The authors have adequately addressed all my earlier concerns and remarks. I think this is a great paper.